# A RAD APPROACH TO DEEP MIXTURE MODELS

**Laurent Dinh**
Google Brain
laurentdinh@google.com

**Jascha Sohl-Dickstein**
Google Brain
jaschasd@google.com

**Razvan Pascanu**
DeepMind
razp@google.com

**Hugo Larochelle**
Google Brain
hugolarochelle@google.com

## ABSTRACT

Flow based models such as REAL NVP are an extremely powerful approach to density estimation. However, existing flow based models are restricted to transforming continuous densities over a continuous input space into similarly continuous distributions over continuous latent variables. This makes them poorly suited for modeling and representing discrete structures in data distributions, for example class membership or discrete symmetries. To address this difficulty, we present a normalizing flow architecture which relies on domain partitioning using locally invertible functions, and possesses both real and discrete valued latent variables. This Real and Discrete (RAD) approach retains the desirable normalizing flow properties of exact sampling, exact inference, and analytically computable probabilities, while at the same time allowing simultaneous modeling of both continuous and discrete structure in a data distribution.

## 1 INTRODUCTION

Latent generative models are one of the prevailing approaches for building expressive and tractable generative models. The generative process for a sample $x$ can be expressed as

$$z \sim p_Z(z)$$
$$x = g(z),$$

where $z$ is a noise vector, and $g$ a parametric *generator network* (typically a deep neural network). This paradigm has several incarnations, including *variational autoencoders* (Kingma & Welling, 2014; Rezende et al., 2014), *generative adversarial networks* (Goodfellow et al., 2014), and *flow based models* (Baird et al., 2005; Tabak & Turner, 2013; Dinh et al., 2015; 2017; Kingma & Dhariwal, 2018; Chen et al., 2018; Grathwohl et al., 2019).

The training process and model architecture for many existing latent generative models, and for all published flow based models, assumes a unimodal smooth distribution over latent variables **z**. Given the parametrization of $g$ as a neural network, the mapping to **x** is a continuous function. This imposed structure makes it challenging to model data distributions with discrete structure – for instance, multi-modal distributions, distributions with holes, distributions with discrete symmetries, or distributions that lie on a union of manifolds (as may approximately be true for natural images, see Tenenbaum et al., 2000). Indeed, such cases require the model to learn a generator whose input Jacobian has highly varying or infinite magnitude to separate the initial noise source into different clusters. Such variations imply a challenging optimization problem due to large changes in curvature. This shortcoming can be critical as several problems of interest are hypothesized to follow a clustering structure, i.e. the distributions is concentrated along several disjoint connected sets (Eghbal-zadeh et al., 2018).

A standard way to address this issue has been to use *mixture models* (Yeung et al., 2017; Richardson & Weiss, 2018; Eghbal-zadeh et al., 2018) or structured priors (Johnson et al., 2016). In order to efficiently parametrize the model, mixture models are often formulated as a *discrete latent variable models* (Hinton & Salakhutdinov, 2006; Courville et al., 2011; Mnih & Gregor, 2014; van den Oord

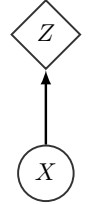
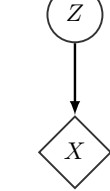
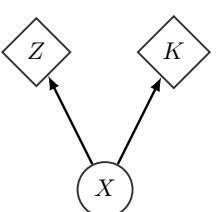
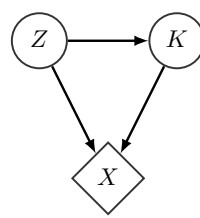

(a) Inference graph for flow based model.

(b) Sampling graph for flow based model.

(c) Inference graph for RAD model.

(d) Sampling graph for RAD model.

Figure 1: Stochastic computational graphs for inference and sampling for flow based models (1a, 1b) and a RAD model (1c, 1d). Note the dependency of $K$ on $Z$ in 1d. While this is not necessary, we will exploit this structure as highlighted later in the main text and in Figure 4.

et al., 2017), some of which can be expressed as a *deep mixture model* (Tang et al., 2012; Van den Oord & Schrauwen, 2014; van den Oord & Dambre, 2015). Although the resulting exponential number of mixture components with depth in deep mixture models is an advantage in terms of expressivity, it is an impediment to inference, evaluation, and training of such models, often requiring as a result the use of approximate methods like *hard*-EM or variational inference (Neal & Hinton, 1998).

In this paper we combine piecewise invertible functions with discrete auxiliary variables, selecting which invertible function applies, to describe a deep mixture model. This framework enables a probabilistic model's latent space to have both real and discrete valued units, and to capture both continuous and discrete structure in the data distribution. It achieves this added capability while preserving the exact inference, exact sampling, exact evaluation of log-likelihood, and efficient training that make standard flow based models desirable.

## 2 MODEL DEFINITION

We aim to learn a parametrized distribution $p_X(\boldsymbol{x})$ on the continuous input domain $\mathbb{R}^d$ by maximizing log-likelihood. The major obstacle to training an expressive probabilistic model is typically efficiently evaluating log-likelihood.

### 2.1 PARTITIONING

If we consider a mixture model with a large number $|K|$ of components, the likelihood takes the form

$$p_X(\boldsymbol{x}) = \sum_{k=1}^{|K|} p_K(k) p_{X|K}(\boldsymbol{x} \mid k).$$

In general, evaluating the likelihood requires computing probabilities for all $|K|$ components. However, following a strategy similar to Rainforth et al. (2018), if we partition the domain $\mathbb{R}^d$ into disjoint subsets $\mathbb{A}_k$ for $1 \le k \le |K|$ such that $\forall i \ne j \; \mathbb{A}_i \cap \mathbb{A}_j = \varnothing$ and $\bigcup_{k=1}^{|K|} \mathbb{A}_k = \mathbb{R}^d$, constrain the support of $p_{X|K}(\boldsymbol{x} \mid k)$ to $\mathbb{A}_k$ (i.e. $\forall \boldsymbol{x} \notin \mathbb{A}_k, p_{X|K}(\boldsymbol{x} \mid k) = 0$), and define a set identification function $f_K(\boldsymbol{x})$ such that $\forall \boldsymbol{x} \in \mathbb{R}^d, \boldsymbol{x} \in \mathbb{A}_{f_K(\boldsymbol{x})}$, we can write the likelihood as

$$p_X(\boldsymbol{x}) = p_K\big(f_K(\boldsymbol{x})\big) p_{X|K}\big(\boldsymbol{x} \mid f_K(\boldsymbol{x})\big).$$

This transforms the problem of summation to a search problem $\boldsymbol{x} \mapsto f_K(\boldsymbol{x})$. This can be seen as the inferential converse of a *stratified sampling* strategy (Rubinstein & Kroese, 2016).

### 2.2 CHANGE OF VARIABLE FORMULA

The proposed approach will be a direct extension of flow based models (Rippel & Adams, 2013; Dinh et al., 2015; 2017; Kingma & Dhariwal, 2018). Flow based models enable log-likelihood

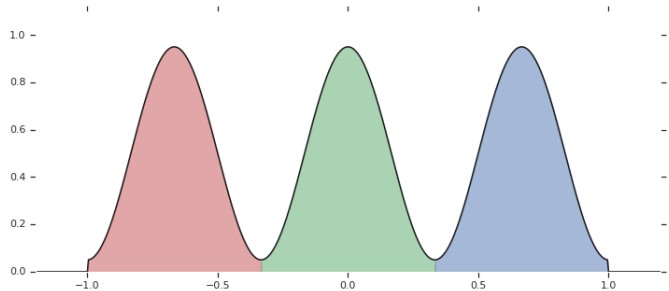

(a) An example of a trimodal distribution $p_X$, sinusoidal distribution. The different modes are colored in red, green, and blue.

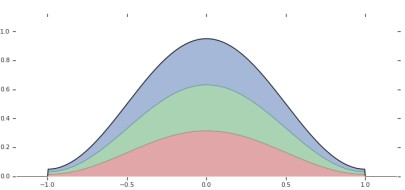

(b) The resulting unimodal distribution $p_Z$, corresponding to the distribution of any of the initial modes in $p_X$.

(c) An example $f_Z(x)$ of a piecewise invertible function aiming at transforming $p_Z$ into a unimodal distribution. The red, green, and blue zones corresponds to the different modes in input space.

Figure 2: Example of a trimodal distribution (2a) turned into a unimodal distribution (2b) using a piecewise invertible function (2c). Note that the initial distribution $p_X$ correspond to an unfolding of $p_{f_Z(X)}$ as $p_X(x) = \frac{1}{3}\big(p_Z(-3x) + p_Z(3x - \frac{2}{3}) + p_Z(3x + \frac{2}{3})\big)$.

evaluation by relying on the *change of variable formula*

$$p_X(\boldsymbol{x}) = p_Z\big(f_Z(\boldsymbol{x})\big)\left|\frac{\partial f_Z}{\partial \boldsymbol{x}^T}(\boldsymbol{x})\right|,$$

with $f_Z$ a parametrized bijective function from $\mathbb{R}^d$ onto $\mathbb{R}^d$ and $\left|\frac{\partial f_Z}{\partial \boldsymbol{x}^T}\right|$ the absolute value of the determinant of its Jacobian.

As also proposed in Falorsi et al. (2019), we relax the constraint that $f_Z$ be bijective, and instead have it be surjective onto $\mathbb{R}^d$ and piecewise invertible. That is, we require $f_{Z|\mathbb{A}_k}(\boldsymbol{x})$ be an invertible function, where $f_{Z|\mathbb{A}_k}(\boldsymbol{x})$ indicates $f_Z(\boldsymbol{x})$ restricted to the domain $\mathbb{A}_k$. Given a distribution $p_{Z,K}(\boldsymbol{z}, k) = p_{K|Z}(k \mid \boldsymbol{z})\, p_Z(\boldsymbol{z})$ such that $\forall(\boldsymbol{z}, k), \boldsymbol{z} \notin f_Z(\mathbb{A}_k) \Rightarrow p_{Z,K} = 0$, we can define the following generative process:

$$\boldsymbol{z}, k \sim p_{Z,K}(\boldsymbol{z}, k)$$
$$\boldsymbol{x} = (f_{Z|\mathbb{A}_k})^{-1}(\boldsymbol{z}).$$

If we use the set identification function $f_K$ associated with $\mathbb{A}_k$, the distribution corresponding to this stochastic inversion can be defined by a change of variable formula

$$p_X(\boldsymbol{x}) = \sum_{k=1}^{|K|} p_{Z,K}\big(f_Z(\boldsymbol{x}), k\big)\left|\frac{\partial f_{Z|\mathbb{A}_k}}{\partial \boldsymbol{x}^T}\right|$$
$$= p_{Z,K}\big(f_Z(\boldsymbol{x}), f_K(\boldsymbol{x})\big)\left|\frac{\partial f_Z}{\partial \boldsymbol{x}^T}\right|.$$

Because of the use of both *Real and Discrete* stochastic variables, we call this class of model RAD. The particular parametrization we use on is depicted in Figure 2. We rely on piecewise invertible functions that allow us to define a mixture model of repeated symmetrical patterns, following a

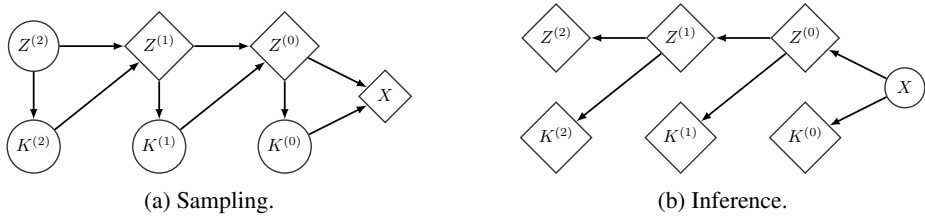

(a) Sampling.  (b) Inference.

Figure 3: Stochastic computational graph in a deep RAD mixture model of $\prod_{l=1}^{3} |K^{(l)}|$ components.

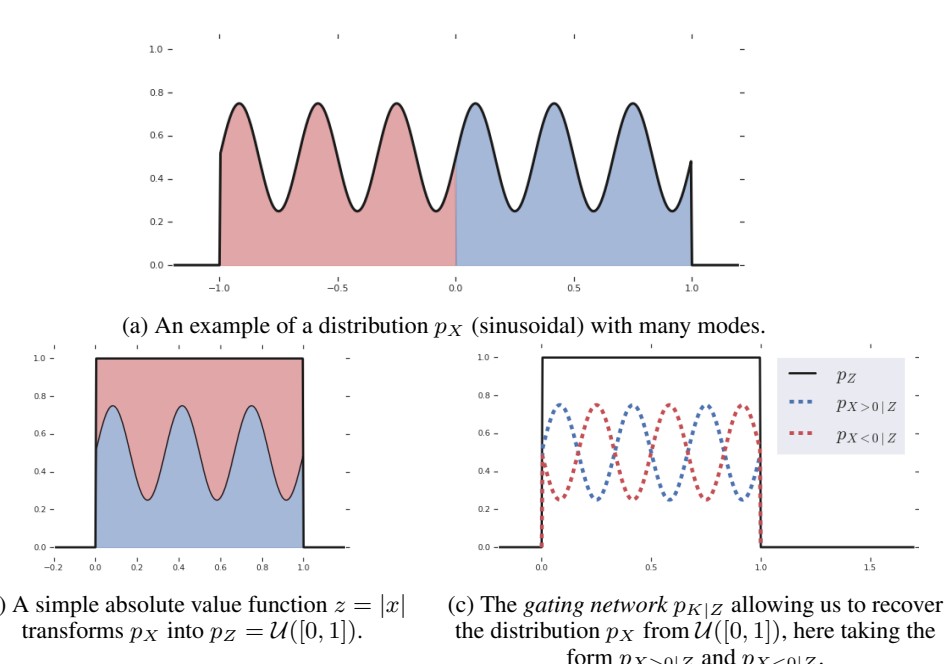

(a) An example of a distribution $p_X$ (sinusoidal) with many modes.

(b) A simple absolute value function $z = |x|$ transforms $p_X$ into $p_Z = \mathcal{U}([0,1])$.

(c) The *gating network* $p_{K|Z}$ allowing us to recover the distribution $p_X$ from $\mathcal{U}([0,1])$, here taking the form $p_{X>0|Z}$ and $p_{X<0|Z}$.

Figure 4: Illustration of the expressive power the gating distribution $p_{K|Z}$ provides. By capturing the structure in a sine wave in $p_{K|Z}$, the function $z, k \mapsto x$ can take on an extremely simple form, corresponding only to a linear function with respect to $z$.

method of *folding the input space*. Note that in this instance the function $f_K$ is implicitly defined by $f_Z$, as the discrete latent corresponds to which invertible component of the piecewise function $x$ falls on.

So far, we have defined a mixture of $|K|$ components with disjoint support. However, if we factorize $p_{Z,K}$ as $p_Z \cdot p_{K|Z}$, we can apply another piecewise invertible map to $Z$ to define $p_Z$ as another mixture model. Recursively applying this method results in a deep mixture model (see Figure 3).

Another advantage of such factorization is in the *gating network* $p_{K|Z}$, as also designated in (van den Oord & Dambre, 2015). It provides a more constrained but less sample wasteful approach than rejection sampling (Bauer & Mnih, 2019) by taking into account the untransformed sample $\boldsymbol{z}$ before selecting the mixture component $k$. This allows the model to exploit the distribution $p_Z$ in different regions $\mathbb{A}_k$ in more complex ways than repeating it as a patternm as illustrated in Figure 4.

The function from the input to the discrete variables, $f_K(\boldsymbol{x})$, contains discontinuities. This presents the danger of introducing discontinuities into $\log p_X(\boldsymbol{x})$, making optimization more difficult. However, by carefully imposing boundary conditions on the gating network, we are able to exactly counteract the effect of discontinuities in $f_K$, and cause $\log p_X(\boldsymbol{x})$ to remain continuous with respect to the parameters. This is discussed in detail in Appendix A.

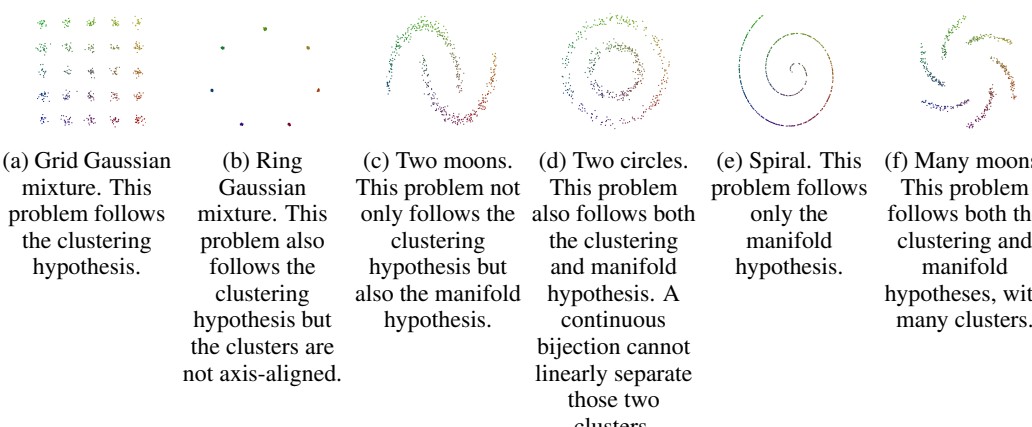

(a) Grid Gaussian mixture. This problem follows the clustering hypothesis.

(b) Ring Gaussian mixture. This problem also follows the clustering hypothesis but the clusters are not axis-aligned.

(c) Two moons. This problem not only follows the clustering hypothesis but also the manifold hypothesis.

(d) Two circles. This problem also follows both the clustering and manifold hypothesis. A continuous bijection cannot linearly separate those two clusters.

(e) Spiral. This problem follows only the manifold hypothesis.

(f) Many moons. This problem follows both the clustering and manifold hypotheses, with many clusters.

Figure 5: Samples drawn from the data distribution in each of several toy two dimensional problems.

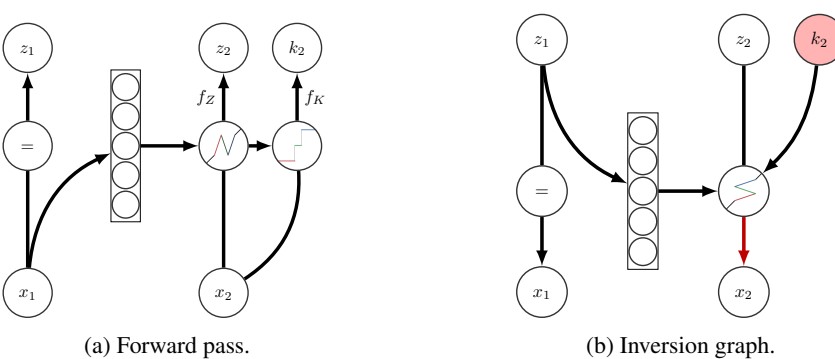

(a) Forward pass.

(b) Inversion graph.

Figure 6: Computational graph of the coupling layers used in the experiments.

# 3 EXPERIMENTS

## 3.1 PROBLEMS

We conduct a brief comparison on six two-dimensional toy problems with REAL NVP to demonstrate the potential gain in expressivity RAD models can enable. Synthetic datasets of $10,000$ points each are constructed following the *manifold hypothesis* and/or the *clustering hypothesis*. We designate these problems as: *grid Gaussian mixture*, *ring Gaussian mixture*, *two moons*, *two circles*, *spiral*, and *many moons* (see Figure 5).

## 3.2 ARCHITECTURE

For the RAD model implementation, we use the piecewise linear activations defined in Appendix A in a coupling layer architecture (Dinh et al., 2015; 2017) for $f_Z$ where, instead of a conditional linear transformation, the conditioning variable $x_1$ determines the parameters of the piecewise linear activation on $x_2$ to obtain $z_2$ and $k_2$, with $z_1 = x_1$ (see Figure 6). For the gating network $p_{K|Z}$, the gating logit neural network $s(z)$ take as input $z = (z_1, z_2)$. We compare with a REAL NVP model using only affine coupling layers. $p_Z$ is a standard Gaussian distribution.

As both these models can easily approximately solve these generative modeling tasks provided enough capacity, we study these model in a relatively low capacity regime, where we can showcase the potential expressivity RAD may provide. Each of these models uses six coupling layers, and each coupling layer uses a one-hidden-layer rectified network with a tanh output activation scaled by a scalar parameter as described in Dinh et al. (2017). For RAD, the logit network $s(\cdot)$ also uses a one-hidden-layer rectified neural network, but with linear output. In order to fairly compare with respect to number of parameters, we provide REAL NVP seven times more hidden units per

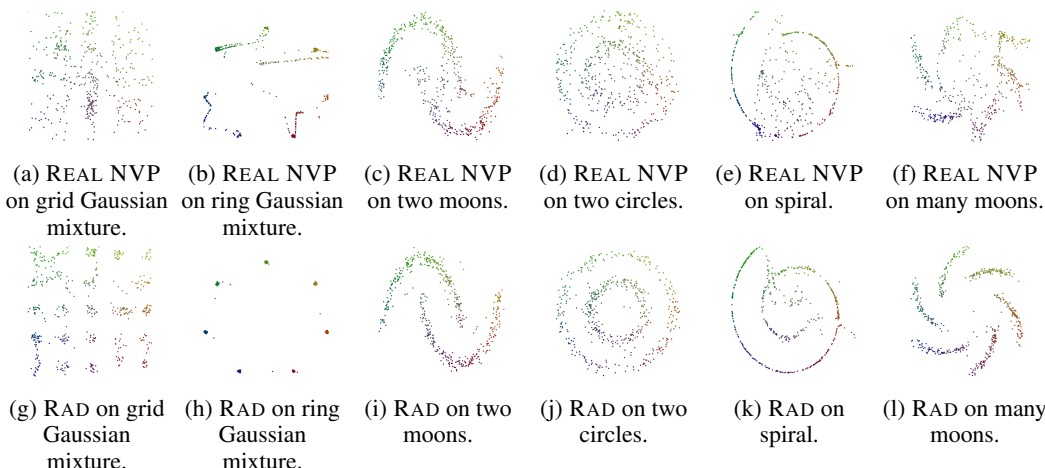

(a) REAL NVP on grid Gaussian mixture.    (b) REAL NVP on ring Gaussian mixture.    (c) REAL NVP on two moons.    (d) REAL NVP on two circles.    (e) REAL NVP on spiral.    (f) REAL NVP on many moons.

(g) RAD on grid Gaussian mixture.    (h) RAD on ring Gaussian mixture.    (i) RAD on two moons.    (j) RAD on two circles.    (k) RAD on spiral.    (l) RAD on many moons.

Figure 7: Comparison of samples from trained REAL NVP (top row) (a-f) and RAD (bottow row) (g-l) models. REAL NVP fails in a low capacity setting by attributing probability mass over spaces where the data distribution has low density. Here, these spaces often connect data clusters, illustrating the challenges that come with modeling multimodal data as one continuous manifold.

hidden layer than RAD, which uses $8$ hidden units per hidden layer. For each level, $p_{K|Z}$ and $f_Z$ are trained using stochastic gradient ascent with ADAM (Kingma & Ba, 2015) on the log-likelihood with a batch size of $500$ for $50,000$ steps.

## 3.3 RESULTS

In each of these problems, RAD is consistently able to obtain higher log-likelihood than REAL NVP.

|  | RAD | REAL NVP |
|---|---|---|
| Grid Gaussian mixture | $-1.20$ | $-2.26$ |
| Ring Gaussian mixture | $3.57$ | $1.85$ |
| Two moons | $-1.21$ | $-1.48$ |
| Two cicles | $-1.81$ | $-2.17$ |
| Spiral | $0.29$ | $-0.36$ |
| Many moons | $-0.83$ | $-1.50$ |

### 3.3.1 SAMPLING AND GAUSSIANIZATION

We plot the samples (Figure 7) of the described RAD and REAL NVP models trained on these problems. In the described low capacity regime, REAL NVP fails by attributing probability mass over spaces where the data distribution has low density. This is consistent with the *mode covering* behavior of maximum likelihood. However, the particular inductive bias of REAL NVP is to prefer modeling the data as one connected manifold. This results in the unwanted probability mass being distributed along the space between clusters.

Flow-based models often follow the principle of *Gaussianization* (Chen & Gopinath, 2001), i.e. transforming the data distribution into a Gaussian. The inversion of that process on a Gaussian distribution would therefore approximate the data distribution. We plot in Figure 8 the inferred Gaussianized variables $z^{(5)}$ for both models trained on the ring Gaussian mixture problem. The Gaussianization from REAL NVP leaves some area of the standard Gaussian distribution unpopulated. These unattended areas correspond to unwanted regions of probability mass in the input space. RAD suffers significantly less from this problem.

An interesting feature is that RAD seems also to outperform REAL NVP on the spiral dataset. One hypothesis is that the model successfully exploits some non-linear symmetries in this problem.

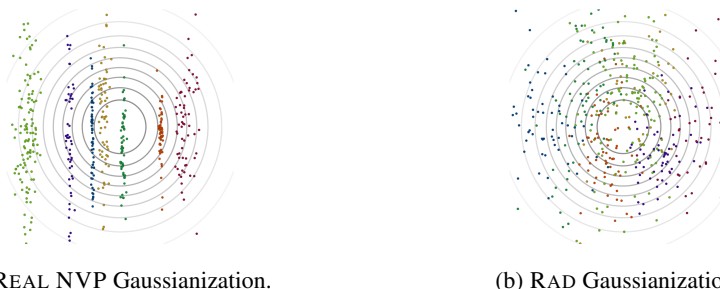

(a) REAL NVP Gaussianization.      (b) RAD Gaussianization.

Figure 8: Comparison of the Gaussianization process for RAD and REAL NVP on the ring Gaussian mixture problem. Both plots show the image of data samples in the latent $z$ variables, with level sets of the standard normal distribution plotted for reference. REAL NVP leaves some area of this Gaussian unpopulated, an effect which is not visually apparent for RAD.

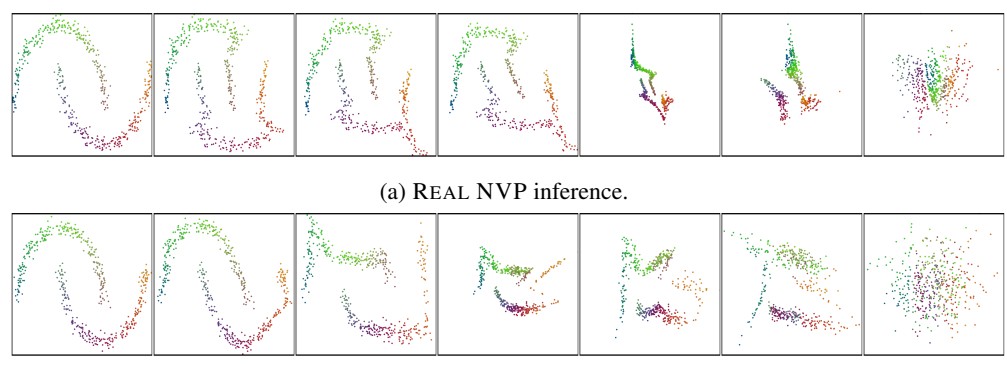

(a) REAL NVP inference.

(b) RAD inference.

Figure 9: Comparison of the inference process for RAD and REAL NVP on the two moons problem. Each pane shows input samples embedded in different networks layers, progressing from left to right from earlier to later network layers. The points are colored according to their original position in the input space. In RAD several points which were far apart in the input space become neighbors in $z^{(5)}$. This is not the case for REAL NVP.

### 3.3.2 FOLDING

We take a deeper look at the Gaussianization process involved in both models. In Figure 9 we plot the inference process of $z^{(5)}$ from $x$ for both models trained on the two moons problem. As a result of a folding process similar to that in Montufar et al. (2014), several points which were far apart in the input space become neighbors in $z^{(5)}$ in the case of RAD.

We further explore this folding process using the visualization described in Figure 10. We verify that the non-linear folding process induced by RAD plays at least two roles: bridging gaps in the distribution of probability mass, and exploiting symmetries in the data.

We observe that in the case of the ring Gaussian mixture (Figure 11a), RAD effectively uses foldings in order to bridge the different modes of the distribution into a single mode, primarily in the last layers of the transformation. We contrast this with REAL NVP (Figure 11b) which struggles to combine these modes under the standard Gaussian distribution using bijections.

In the spiral problem (Figure 12), RAD decomposes the spiral into three different lines to bridge (Figure 12a) instead of unrolling the manifold fully, which REAL NVP struggles to do (Figure 12b).

In both cases, the points remain generally well separated by labels, even after being pushed through a RAD layer (Figure 11a and 12a). This enables the model to maximize the conditional log-probability $\log(p_{K|Z})$.

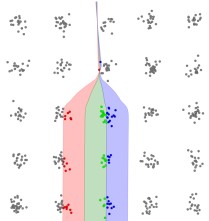
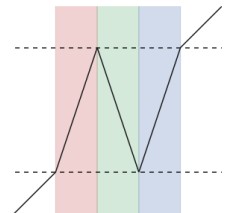
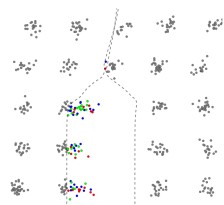
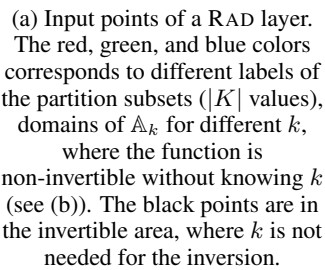

(a) Input points of a RAD layer. The red, green, and blue colors corresponds to different labels of the partition subsets ($|K|$ values), domains of $\mathbb{A}_k$ for different $k$, where the function is non-invertible without knowing $k$ (see (b)). The black points are in the invertible area, where $k$ is not needed for the inversion.

(b) An example of piecewise linear function used in a RAD layer. The red, green, and blue colors corresponds to the different labels of the partition subsets in the non-invertible area. The dashed lines correspond to the non-invertible area in output space.

(c) Output points of a RAD layer. The red, green, and blue colors corresponds to the different labels of the partition subsets in the non-invertible area of the input space, where points are folded on top of each other. The black points are in the invertible area, where $k$ is not needed for the inversion. The dashed lines correspond to the non-invertible area in output space.

Figure 10: Understanding the folding process, and understanding other visualizations of the folding process.

## 4 CONCLUSION

We introduced an approach to tractably evaluate and train deep mixture models using piecewise invertible maps as a folding mechanism. This allows exact inference, exact generation, and exact evaluation of log-likelihood, avoiding many issues in previous discrete variables models. This method can easily be combined with other flow based architectural components, allowing flow based models to better model datasets with discrete as well as continuous structure.

### ACKNOWLEDGEMENTS

The authors would like to thank Kyle Kastner, Johanna Hansen, Harm De Vries, Ben Poole, Prajit Ramachandran, Dustin Tran, David Grangier, George J. Tucker, Matt D. Hoffman, Daniel Duckworth, Anna Huang, Fabian Pedregosa, Arvind Neelakantan, Dale Schuurmans, Graham Taylor, Bart van Merrienböer, Daniel Duckworth, Vincent Dumoulin, Marc G. Bellemare, and Ross Goroshin for valuable discussion and feedbacks.

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

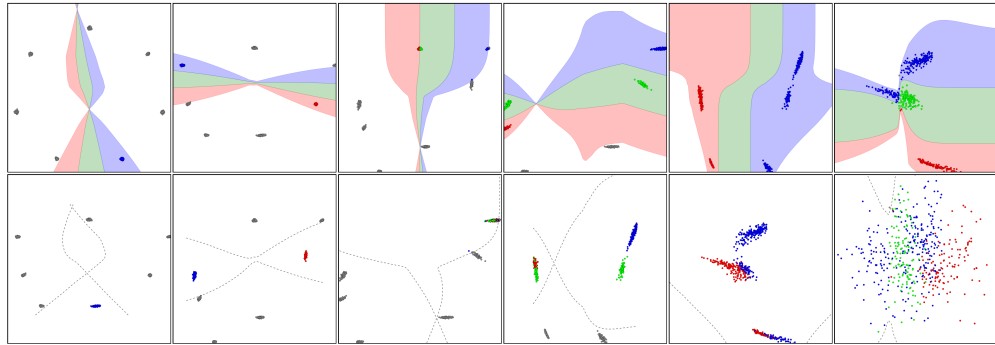

(a) RAD folding strategy on the ring Gaussian mixture problem. The top rows correspond to each RAD layer's input points, and the bottom rows to its output points, as shown in 10. The labels tends to be well separated in output space as well.

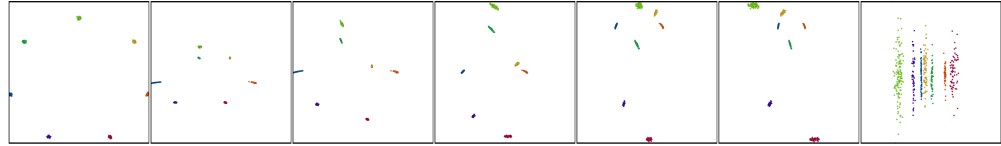

(b) REAL NVP inference strategy on the ring Gaussian mixture problem. The points are colored according to their original position in the input space.

Figure 11: RAD and REAL NVP inference processes on the ring Gaussian mixture problem. Each column correspond to a RAD or affine coupling layer. RAD effectively uses foldings in order to bridge the multiple modes of the distribution into a single mode, primarily in the last layers of the transformation, whereas REAL NVP struggles to bring together these modes under the standard Gaussian distribution using continuous bijections.

Laurent Dinh, Jascha Sohl-Dickstein, and Samy Bengio. Density estimation using real nvp. In *International Conference on Learning Representations*, 2017.

Hamid Eghbal-zadeh, Werner Zellinger, and Gerhard Widmer. Mixture density generative adversarial networks. *Neural Information Processing Systems: Bayesian Deep Learning Workshop*, 2018.

Luca Falorsi, Pim de Haan, Tim R. Davidson, and Patrick Forr. Reparameterizing distributions on lie groups. In *Proceedings of the twenty-second international conference on artificial intelligence and statistics*, 2019.

Ian Goodfellow, Jean Pouget-Abadie, Mehdi Mirza, Bing Xu, David Warde-Farley, Sherjil Ozair, Aaron Courville, and Yoshua Bengio. Generative adversarial nets. In *Advances in Neural Information Processing Systems*, pp. 2672–2680, 2014.

Will Grathwohl, Dami Choi, Yuhuai Wu, Geoffrey Roeder, and David Duvenaud. Backpropagation through the void: Optimizing control variates for black-box gradient estimation. In *International Conference on Learning Representations*, 2018.

Will Grathwohl, Ricky TQ Chen, Jesse Betterncourt, Ilya Sutskever, and David Duvenaud. Ffjord: Free-form continuous dynamics for scalable reversible generative models. 2019.

Geoffrey E Hinton and Ruslan R Salakhutdinov. Reducing the dimensionality of data with neural networks. *science*, 313(5786):504–507, 2006.

Eric Jang, Shixiang Gu, and Ben Poole. Categorical reparameterization with gumbel-softmax. In *International Conference on Learning Representations*, 2017.

Matthew Johnson, David K Duvenaud, Alex Wiltschko, Ryan P Adams, and Sandeep R Datta. Composing graphical models with neural networks for structured representations and fast inference. In *Advances in neural information processing systems*, pp. 2946–2954, 2016.

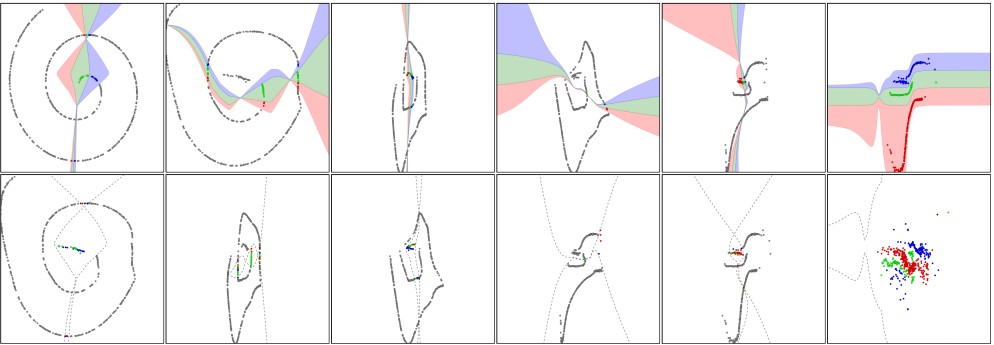

(a) RAD folding strategy on the spiral problem. The top rows correspond to each RAD layer's input points, and the bottom rows to its output points, as shown in 10.

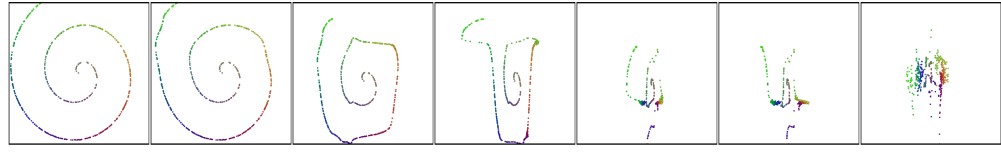

(b) REAL NVP inference strategy on the spiral problem. The points are colored according to their original position in the input space.

Figure 12: RAD and REAL NVP inference processes on the spiral problem. Each column correspond to a RAD or affine coupling layer. Instead of unrolling the manifold as REAL NVP tries to, RAD uses a more successful strategy of decomposing the spiral into three different lines that it later bridges.

Diederik P Kingma and Jimmy Ba. Adam: A method for stochastic optimization. In *International Conference on Learning Representations*, 2015.

Diederik P Kingma and Max Welling. Auto-encoding variational bayes. In *International Conference on Learning Representations*, 2014.

Durk P Kingma and Prafulla Dhariwal. Glow: Generative flow with invertible 1x1 convolutions. In *Advances in Neural Information Processing Systems*, pp. 10236–10245, 2018.

Chris J Maddison, Andriy Mnih, and Yee Whye Teh. The concrete distribution: A continuous relaxation of discrete random variables. In *International Conference on Learning Representations*, 2017.

Andriy Mnih and Karol Gregor. Neural variational inference and learning in belief networks. In *International Conference on Machine Learning*, 2014.

Guido F Montufar, Razvan Pascanu, Kyunghyun Cho, and Yoshua Bengio. On the number of linear regions of deep neural networks. In *Advances in neural information processing systems*, pp. 2924–2932, 2014.

Radford M Neal and Geoffrey E Hinton. A view of the em algorithm that justifies incremental, sparse, and other variants. In *Learning in graphical models*, pp. 355–368. Springer, 1998.

Tom Rainforth, Yuan Zhou, Xiaoyu Lu, Yee Whye Teh, Frank Wood, Hongseok Yang, and Jan-Willem van de Meent. Inference trees: Adaptive inference with exploration. *arXiv preprint arXiv:1806.09550*, 2018.

Danilo Jimenez Rezende, Shakir Mohamed, and Daan Wierstra. Stochastic backpropagation and approximate inference in deep generative models. In *International Conference on Machine Learning*, 2014.

Eitan Richardson and Yair Weiss. On gans and gmms. In *Advances in Neural Information Processing Systems*, pp. 5852–5863, 2018.

Oren Rippel and Ryan Prescott Adams. High-dimensional probability estimation with deep density models. *arXiv preprint arXiv:1302.5125*, 2013.

Jason Tyler Rolfe. Discrete variational autoencoders. In *International Conference on Learning Representations*, 2017.

Reuven Y Rubinstein and Dirk P Kroese. *Simulation and the Monte Carlo method*, volume 10. John Wiley & Sons, 2016.

EG Tabak and Cristina V Turner. A family of nonparametric density estimation algorithms. *Communications on Pure and Applied Mathematics*, 66(2):145–164, 2013.

Yichuan Tang, Ruslan Salakhutdinov, and Geoffrey Hinton. Deep mixtures of factor analysers. In *International Conference on Machine Learning*, 2012.

Joshua B Tenenbaum, Vin De Silva, and John C Langford. A global geometric framework for nonlinear dimensionality reduction. *science*, 290(5500):2319–2323, 2000.

George Tucker, Andriy Mnih, Chris J Maddison, John Lawson, and Jascha Sohl-Dickstein. Rebar: Low-variance, unbiased gradient estimates for discrete latent variable models. In *Advances in Neural Information Processing Systems*, pp. 2627–2636, 2017.

Aäron van den Oord and Joni Dambre. Locally-connected transformations for deep gmms. In *International Conference on Machine Learning (ICML): Deep Learning Workshop*, pp. 1–8, 2015.

Aaron Van den Oord and Benjamin Schrauwen. Factoring variations in natural images with deep gaussian mixture models. In *Advances in Neural Information Processing Systems*, pp. 3518–3526, 2014.

Aaron van den Oord, Oriol Vinyals, et al. Neural discrete representation learning. In *Advances in Neural Information Processing Systems*, pp. 6306–6315, 2017.

Serena Yeung, Anitha Kannan, Yann Dauphin, and Li Fei-Fei. Tackling over-pruning in variational autoencoders. *arXiv preprint arXiv:1706.03643*, 2017.

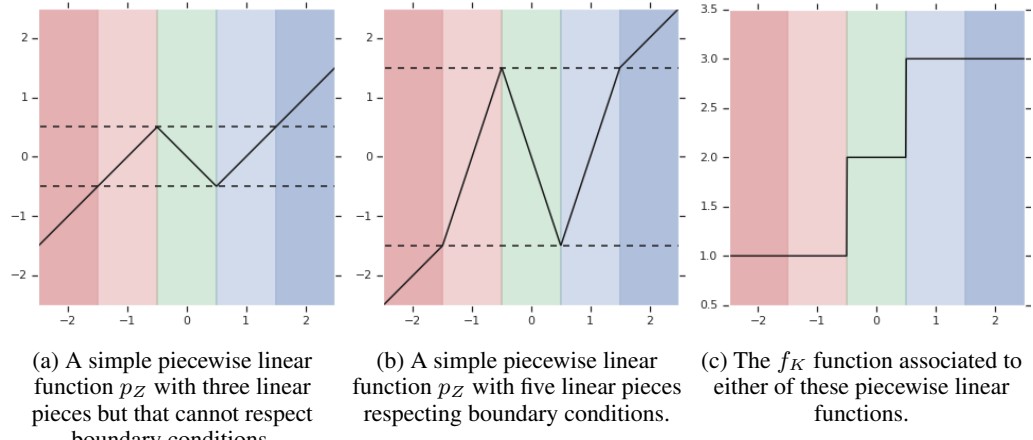

(a) A simple piecewise linear function $p_Z$ with three linear pieces but that cannot respect boundary conditions.

(b) A simple piecewise linear function $p_Z$ with five linear pieces respecting boundary conditions.

(c) The $f_K$ function associated to either of these piecewise linear functions.

Figure 13: Simple piecewise linear scalar function $f_Z$ before 13a and after 13b respecting boundary conditions. The colored area correspond to the different indices for the mixture components, lighter color for non-invertible areas. The dashed line correspond to the non-invertible area in the output space. In 13c, we show the $f_K$ function resulting from these nonlinearities.

## A    CONTINUITY

The standard approach in learning a deep probabilistic model has been stochastic gradient descent on the negative log-likelihood. Although the model enables the computation of a gradient almost everywhere, the log-likelihood is unfortunately discontinuous. Let's decompose the log-likelihood

$$\log\big(p_X(\boldsymbol{x})\big) = \log\big(p_Z\big(f_Z(\boldsymbol{x})\big)\big) + \log\big(p_{K|Z}\big(f_K(\boldsymbol{x}) \mid f_Z(\boldsymbol{x})\big)\big) + \log\left(\left|\frac{\partial f_Z}{\partial \boldsymbol{x}^T}\right|(\boldsymbol{x})\right).$$

There are two sources of discontinuity in this expression: $f_K$ is a function with discrete values (therefore discontinuous) and $\frac{\partial f_Z}{\partial \boldsymbol{x}^T}$ is discontinuous because of the transition between the subsets $\mathbb{A}_k$, leading to the expression of interest

$$\log\big(p_{K|Z}\big(f_K(\boldsymbol{x}) \mid f_Z(\boldsymbol{x})\big)\big) + \log\left(\left|\frac{\partial f_Z}{\partial \boldsymbol{x}^T}\right|(\boldsymbol{x})\right),$$

which takes a role similar to the log-Jacobian determinant, a *pseudo log-Jacobian determinant*.

Let's build from now on the simple scalar case and a piecewise linear function

$$f_Z(x) = \begin{cases} \alpha_1(x + \beta) + \alpha_2\beta, & \text{if } x \le -\beta \\ \alpha_3(x - \beta) - \alpha_2\beta, & \text{if } x \ge \beta \\ -\alpha_2 x, & \text{if } |x| \le \beta \end{cases},$$

or $f_Z(x) = \alpha_1 \min(x + \beta, 0) + \alpha_3 \max(x - \beta, 0) - \alpha_2 \min(\max(x, -\beta), \beta)$ (with $\alpha_k > 0$ and $\beta >= 0$, see figure13a), then $\mathbb{A}_1 =] - \infty, -\beta[, \mathbb{A}_2 = [-\beta, \beta], \mathbb{A}_3 =]\beta, +\infty[$ and $f_K(x) = \mathbb{1}(x > -\beta) + \mathbb{1}(x > \beta)$.

In this case, $s(z) = \big(\log\big(p_{K|Z}\big(k \mid z\big)\big)\big)_{k \le N} + C\mathbb{1}_{|K|}$ can be seen as a vector valued function. We can attempt at parametrizing the model such that the pseudo log-Jacobian determinant becomes continuous with respect to $\beta$ by expressing the boundary condition at $x = \beta$

$$\log\big(p_{K|Z}\big(f_K(\beta^-) \mid f_Z(\beta^-)\big)\big) + \log\big(|f_Z'|(\beta^-)\big)$$
$$= \log\big(p_{K|Z}\big(f_K(\beta^+) \mid f_Z(\beta^+)\big)\big) + \log\big(|f_Z'|(\beta^+)\big)$$
$$\Rightarrow s(-\alpha_2\beta)_2 + \log(\alpha_2) = s(-\alpha_2\beta)_3 + \log(\alpha_3).$$

If we define $\Omega_{2,3} = \begin{bmatrix} 0 & 0 \\ 0 & \Omega \end{bmatrix}$ and $\Omega_{1,2} = \begin{bmatrix} \Omega & 0 \\ 0 & 0 \end{bmatrix}$, with $\Omega = \frac{1}{2}\begin{bmatrix} -1 & 1 \\ 1 & -1 \end{bmatrix}$, then this boundary condition can be enforced, together with a similar one at $x = -\beta$, by replacing the

function $s$ with

$$\beta s(z) + \frac{\beta}{2}\left(1 + cos\left(\frac{(z\alpha_2^{-1} - \beta)\pi}{2\beta}\right)\right) s(z) \cdot \Omega_{2,3}$$

$$-\left(\log(\alpha_1), \log(\alpha_2), \log(\alpha_3)\right) + \frac{\beta}{2}\left(1 + cos\left(\frac{(z\alpha_2^{-1} + \beta)\pi}{2\beta}\right)\right) s(z) \cdot \Omega_{1,2}.$$

Another type of boundary condition can be found at between the non-invertible area and the invertible area $z = \alpha_2\beta$, as $\forall z > \alpha_2\beta, p_{3|Z}(3 \mid z) = 1$, therefore

$$\log\left(p_{K|Z}\left(3 \mid f_Z((1 + \alpha_3^{-1}\alpha_2)\beta))\right)\right) + \log\left(\left|f_Z'\right|\left(\left((1 + \alpha_3^{-1}\alpha_2)\beta\right)^{-}\right)\right)$$
$$= \log\left(\left|f_Z'\right|\left(\left((1 + \alpha_3^{-1}\alpha_2)\beta\right)^{+}\right)\right).$$

Since the condition $\forall k < 3, p_{K|Z}\left(k \mid z\right) \to 0$ when $z \to (\alpha_2\beta)^{-}$ will lead to an infinite loss barrier at $x = -\beta$, another way to enforce this boundary condition is by adding linear pieces (Figure 13b):

$$f_Z(x) = \begin{cases} \alpha_1(x + \beta) + \alpha_2\beta, & \text{if } x \in [-(1 + \alpha_1^{-1}\alpha_2)\beta, -\beta] \\ \alpha_3(x - \beta) - \alpha_2\beta, & \text{if } x \in [\beta, (1 + \alpha_3^{-1}\alpha_2)\beta] \\ \alpha_1 \cdot p_{K|Z}(1 \mid -\alpha_2\beta)\left(x + (1 + \alpha_1^{-1}\alpha_2)\beta\right) - \alpha_2\beta, & \text{if } x < -(1 + \alpha_1^{-1}\alpha_2)\beta \\ \alpha_3 \cdot p_{K|Z}(3 \mid \alpha_2\beta)\left(x - (1 + \alpha_3^{-1}\alpha_2)\beta\right) + \alpha_2\beta, & \text{if } x > (1 + \alpha_3^{-1}\alpha_2)\beta \\ -\alpha_2 x, & \text{if } |x| < \beta \end{cases}$$

$$= \alpha_1\min(x + \beta, 0) + \alpha_3\max(x - \beta, 0) - \alpha_2\min(\max(x, -\beta), \beta)$$
$$- \alpha_1\left(1 - p_{K|Z}(1 \mid -\alpha_2\beta)\right)\min\left(x + (1 + \alpha_1^{-1}\alpha_2)\beta, 0\right)$$
$$- \alpha_3\left(1 - p_{K|Z}(3 \mid \alpha_2\beta)\right)\max\left(x - (1 + \alpha_3^{-1}\alpha_2)\beta, 0\right).$$

The inverse is defined as

$$\forall z \in \mathbb{R}, x = \begin{cases} \alpha_1^{-1}\left(p_{K|Z}(1 \mid -\alpha_2\beta)\right)^{-1}(z + \alpha_2\beta) - \beta, & \text{if } z \leq -\alpha_2\beta \\ \alpha_3^{-1}\left(p_{K|Z}(3 \mid \alpha_2\beta)\right)^{-1}(z - \alpha_2\beta) + \beta, & \text{if } z \geq \alpha_2\beta \\ \alpha_1^{-1}(z - \alpha_2\beta) - \beta, & \text{if } x \in \mathbb{A}_1 \\ \alpha_2^{-1}z, & \text{if } x \in \mathbb{A}_2 \\ \alpha_3^{-1}(z + \alpha_2\beta) + \beta, & \text{if } x \in \mathbb{A}_3 \end{cases}.$$

In order to know the values of $s$ at the boundaries $\pm\alpha_2\beta$, we can use the logit function

$$\beta(a_s z + b_s) + \frac{\beta}{2}\left(1 + cos\left(\frac{(z\alpha_2^{-1} - \beta)\pi}{2\beta}\right)\right)(a_s z + b_s) \cdot \Omega_{2,3}$$

$$-\left(\log(\alpha_1), \log(\alpha_2), \log(\alpha_3)\right) + \frac{\beta}{2}\left(1 + cos\left(\frac{(z\alpha_2^{-1} + \beta)\pi}{2\beta}\right)\right)(a_s z + b_s) \cdot \Omega_{1,2}$$

$$+ \frac{\beta}{2}\left(1 + cos\left(\frac{z\pi}{\alpha_2\beta}\right)\right)s(z),$$

where $(a_s, b_s) \in (\mathbb{R}^3)^2$.

Given those constraints, the model can then be reliably learned through gradient descent methods. Note that the resulting tractability of the model results from the fact that the discrete variables $k$ is only interfaced during inference with the distribution $p_{K|Z}$, unlike discrete variational autoencoders approaches (Mnih & Gregor, 2014; van den Oord et al., 2017) where it is fed to a deep neural network. Similar to Rolfe (2017), the learning of discrete variables is achieved by relying on the the continuous component of the model, and, as opposed as other approaches (Jang et al., 2017; Maddison et al., 2017; Grathwohl et al., 2018; Tucker et al., 2017), this gradient signal extracted is exact and closed form.

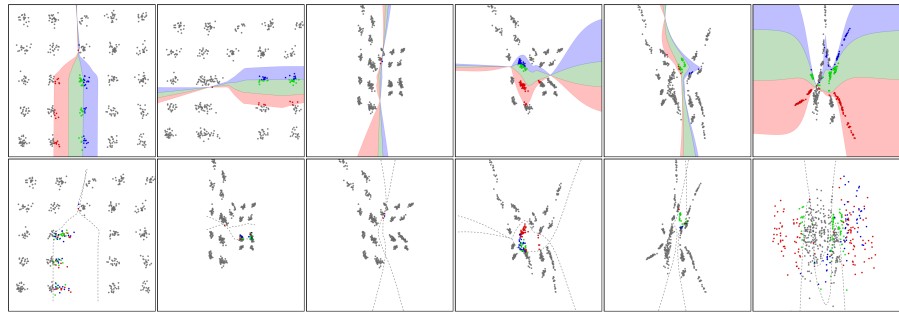

(a) RAD folding strategy on the grid Gaussian mixture problem. The top rows correspond to a RAD layer input points, and the bottom rows to its output points, as shown in 10.

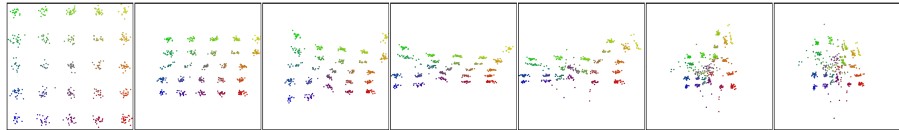

(b) REAL NVP inference strategy on the grid Gaussian mixture problem. The points are colored according to their original position in the input space.

Figure 14: RAD and REAL NVP inference process on the grid Gaussian mixture problem. Each column correspond to a RAD or affine coupling layer.

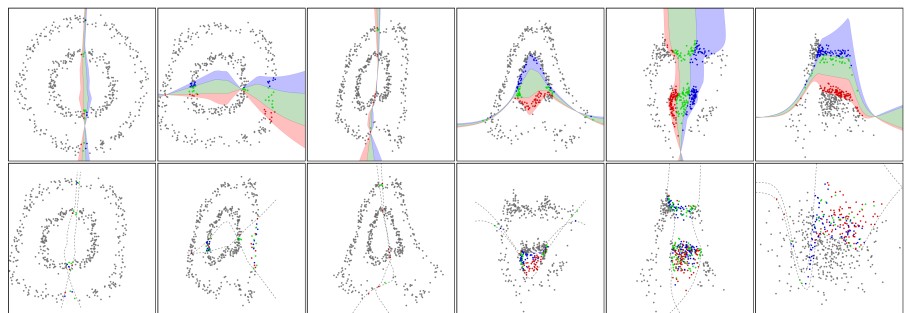

(a) RAD folding strategy on the two circles problem. The top rows correspond to a RAD layer input points, and the bottom rows to its output points, as shown in 10.

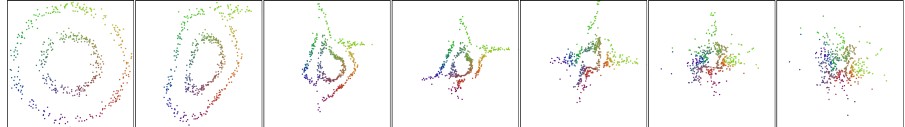

(b) REAL NVP inference strategy on the two circles problem. The points are colored according to their original position in the input space.

Figure 15: RAD and REAL NVP inference process on the two circles problem. Each column correspond to a RAD or affine coupling layer.

## B  INFERENCE PROCESSES

We plot the remaining inference processes of RAD and REAL NVP on the remaining problems not plotted previously: *grid Gaussian mixture* (Figure 14), *two circles* (Figure 15), *two moons* (Figure 16), and *many moons* (Figure 17). We also compare the final results of the Gaussianization processes on both models on the different toy problems in Figure 18.

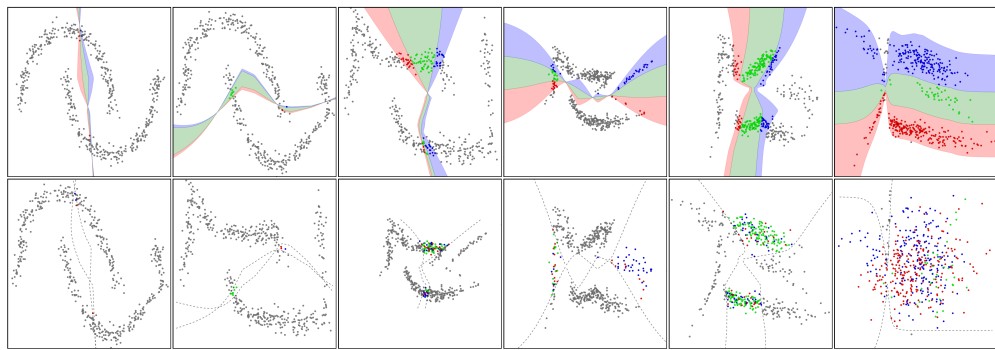

(a) RAD folding strategy on the two moons problem. The top rows correspond to a RAD layer input points, and the bottom rows to its output points, as shown in 10.

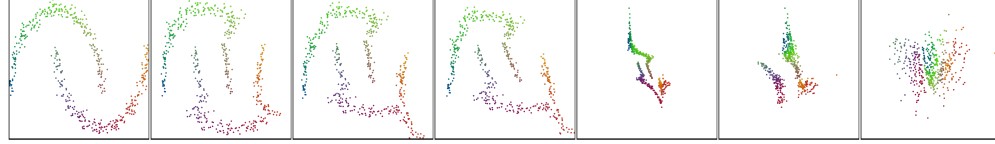

(b) REAL NVP inference strategy on the two moons problem. The points are colored according to their original position in the input space.

Figure 16: RAD and REAL NVP inference process on the two moons problem. Each column correspond to a RAD or affine coupling layer.

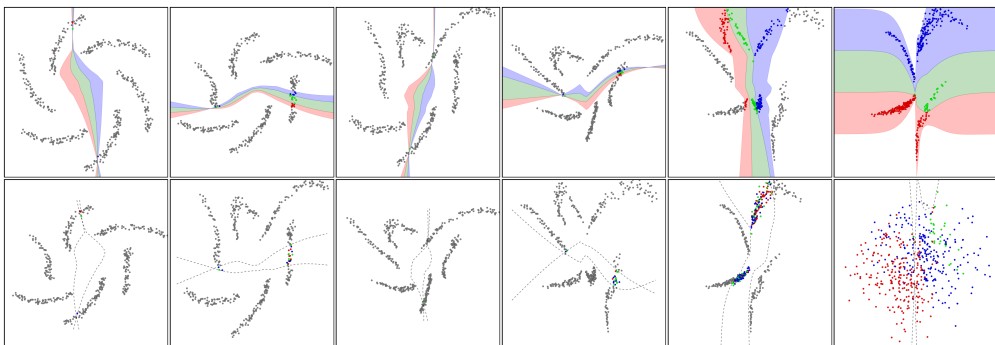

(a) RAD folding strategy on the many moons problem. The top rows correspond to a RAD layer input points, and the bottom rows to its output points, as shown in 10.

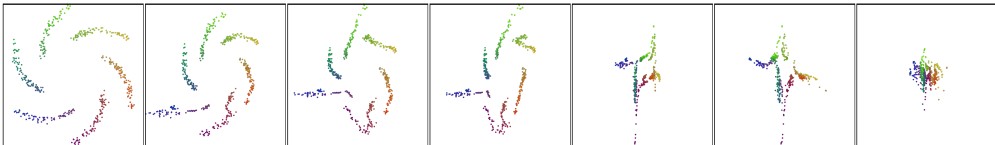

(b) REAL NVP inference strategy on the many moons problem. The points are colored according to their original position in the input space.

Figure 17: RAD and REAL NVP inference process on the many moons problem. Each column correspond to a RAD or affine coupling layer.

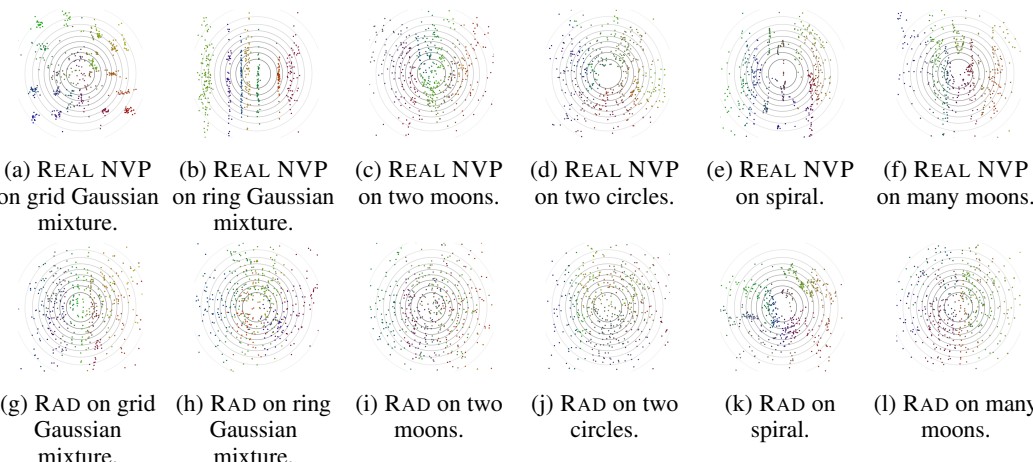

(a) REAL NVP on grid Gaussian mixture.

(b) REAL NVP on ring Gaussian mixture.

(c) REAL NVP on two moons.

(d) REAL NVP on two circles.

(e) REAL NVP on spiral.

(f) REAL NVP on many moons.

(g) RAD on grid Gaussian mixture.

(h) RAD on ring Gaussian mixture.

(i) RAD on two moons.

(j) RAD on two circles.

(k) RAD on spiral.

(l) RAD on many moons.

Figure 18: Comparison of the Gaussianization from the trained REAL NVP (top row) (a-f) and RAD (bottow row) (g-l). REAL NVP fails in a low capacity setting by leaving unpopulated areas where the standard Gaussian attributes probability mass. Here, these spaces as often ones separating clusters, showing the failure in modeling the data as one manifold.

