# OpenReview forum: "A RAD approach to deep mixture models"
_ICLR.cc/2019/Workshop/DeepGenStruct — DeepGenStruct 2019_

### Official Review · AnonReviewer2 · 2019-04-10
**A normalizing flow that uses a partition of the output space into non-overlapping regions and a separate invertible mapping on each region. Promising idea, but I have some practical concerns**

**Rating:** 3
**Confidence:** 2

**Review:**

This paper proposes an invertible transformation, similarly to a normalizing flow, that partitions the domain of a continuous density into non-overlapping regions, and uses a piece-wise invertible mapping for each of these regions.

From the abstract, it may seem that the authors will tackle the case where x is a discrete variable; however the transformation is for distributions continuous support; I suggest to rewrite the abstract to avoid that misunderstanding.

The idea of the paper seems promising. My main concerns are of practical nature. In particular, the method introduces many parameters (the partitioning of the regions, the piece-wise mappings, etc.) that seem hard to tune in practice. Can the authors discuss how (and give the rationale why) to set all of these components their specific choices?

Also, how does the proposed approach scale with dimensionality? It is hard to partition a high-dimensional space into meaningful regions.

Notation: What is K in the paper? It must be a set, since |K| is the number of components, what K isn't defined anywhere in the paper.

Typo: patternm => pattern

---

### Official Review · AnonReviewer1 · 2019-04-18
**A nice paper that uses mixture of flows to capture discrete structure in the data, but lacks convincing experiments on real datasets**

**Rating:** 3
**Confidence:** 3

**Review:**

General:

This paper proposes a new framework to combine mixture models and invertible projections to model discrete structures in the data. This new method adds flexibility to existing flow-based generative models, especially in the case of  modeling distributions with multiple modalities, holes, or clear clustering structures. I think this is a nice paper which tries to relax the limitation of existing flow methods -- it is usually difficult for them to map between two manifolds that have holes, are non-smooth, or have distinct modality patterns.

Pros:

+ Nice idea and clear formulation. The method relaxes the constraint present in existing flow-based models that the entire projection needs to be invertible, instead the projection is only piecewise invertible -- but they are still fully invertible if the membership variable (auxiliary variable) k is given. Through introducing this auxiliary discrete variable, each partition in the data would correspond to a separate locally invertible function. I think this way would greatly improve the flexibility of flow models, relaxing the requirement for high-varying Jacobian term to shape the manifold in some cases (which is often hard to achieve). Also, this method preserves the merits of existing flow-based models (e.g. exact learning/inference)

+ This paper includes a series of very clear visualizations to explain the methods and compare with the RealNVP model. The latent space visualizations at different coupling layers provide insights on how flows transform the space (or how one density space flows to another) step by step. I really enjoy reading these figures.

+ The proposed method outperforms RealNVP in terms of log likelihood on several toy datasets.


Cons:

- It is unknown how this approach would work in real settings. This paper only carried out the experiments on very simple data with limited capacities. There might be some issues when dealing with real dataset. I am a bit concerned about the applicability of this approach to practical problems, I think more thorough experiments need to done to prove its performance.

- How to pick the number of modes K ? Is this a hyperparameter ? If so, the results might be sensitive to it in real settings.

- In Appendix A the authors provide a piecewise linear formulation of f_z(x), is this general enough ? It is unclear to me how to design such functions when modeling real-world image or speech data.

---

### Decision · Program_Chairs · 2019-04-19
**Acceptance Decision**

**Decision:**

Accept

**Comment:**

Accepted